# VPD-100K: Towards Generalizable and Fine-grained Visual Privacy Protection

**Xiaobin Hu** [1]  **Enpu Zuo** [1]  **Lanping Hu** [2]  **Kaiwen Yang** [2]  **Dianshu Liao** [2]  **Tianyi Zhang** [2]  **Bo Yin** [1]  **Yinsi Zhou** [3]  **Shidong Pan** [4]  **Xiaoyu Sun** [2]

## Abstract

Privacy protection has become a critical requirement in the era of ubiquitous visual data sharing, imposing higher demands on efficient and robust privacy detection algorithms. However, current robust detection models are severely hindered by the lack of comprehensive datasets. Existing privacy-oriented datasets often suffer from limited scale, coarse-grained annotations, and narrow domain coverage, failing to capture the intricate details of sensitive information in real-world environments. To bridge this gap, we present a large-scale, fine-grained **V**isual **P**rivacy **D**ataset (VPD-100K), designed to facilitate generalized privacy detection. We establish a holistic taxonomy comprising four primary domains: Human Presence, On-Screen Personally Identifiable Information (PII), Physical Identifiers, and Location Indicators, containing 100,000 images annotated with 33 fine-grained classes and over 190,000 object instances. Statistical analysis reveals that our dataset features long-tailed distributions, small object scales, and high visual complexity. These characteristics make the dataset particularly valuable for demanding, unconstrained applications such as live streaming, where actors frequently face unintentional, real-time information leakage. Furthermore, we design an effective frequency-enhanced lightweight module consisting of frequency-domain attention fusion and adaptive spectral gating mechanism that breaks the limitations of spatial pixel intensity to better capture the subtle details of sensitive information. Extensive experiments conducted on both diverse image and streaming videos

[1]National University of Singapore [2]Australian National University [3]The University of New South Wales [4]New York University. Correspondence to: Xiaobin Hu, Shidong Pan, Xiaoyu Sun <ben0xiaobin0hu1@nus.edu.sg, shidong.pan@nyu.edu, xiaoyu.sun1@anu.edu.au.>.

*Proceedings of the 43$^{rd}$ International Conference on Machine Learning*, Seoul, South Korea. PMLR 306, 2026. Copyright 2026 by the author(s).

benchmarks consistently demonstrate the effectiveness of our VPD-100K dataset and the well-curated frequency mechanism. The code and dataset are available at https://vpd-100k.github.io/.

## 1. Introduction

Privacy protection has become a fundamental requirement in the era of ubiquitous visual data sharing, imposing higher demands on efficient and robust privacy detection algorithms across legal and technical domains (Tao et al., 2025; Pan et al., 2025; Sun et al., 2021a). An ideal detection model must not only identify obvious sensitive targets but also maintain high precision and real-time responsiveness in unconstrained, complex environments (Caliskan Islam et al., 2014). Current privacy detection methods (Chen et al., 2021; Kqiku & Reinhardt, 2024; Xompero et al., 2024) largely follow two paradigms: image-level sensitivity prediction and object-level identifier localization. While the former offers a broad assessment of privacy risks, it lacks the fine-grained localization necessary for effective redaction. Conversely, existing object-level datasets (Zhao et al., 2022), though more precise, are severely hindered by their limited scale, coarse-grained annotations, and narrow domain coverage. For instance, datasets like Privacy Alert (Zhao et al., 2022) and DIPA (Xu et al., 2023) often fail to capture the intricate details of sensitive information, such as on-screen Personally Identifiable Information (PII) or localized environmental markers required for real-world applications like live streaming.

This deficiency is not merely a collection flaw but a symptom of the inherent difficulty in balancing data scale with ethical constraints. Lacking comprehensive and diverse examples (Wen et al., 2024; Meden et al., 2021), current models fail to learn robust privacy priors and are forced to rely on coarse categories as a crutch. This data gap manifests in three key limitations: (1) Limited Scale and Availability: Existing datasets like BIV-Priv (Tseng et al., 2025) and DIPA2 (Xu et al., 2024) contain only a few images, which is insufficient for training large-scale generalized detectors. (2) Coarse Taxonomy: Many works (Zerr et al., 2012) provide only high-level tags (*e.g., "person", "by-*

*stander", "other people"*), failing to distinguish between nuanced privacy levels. (3) Narrow Domain Coverage: Most datasets ignore the most critical leakage source in modern digital life—on-screen PII (*e.g., "email", "password", "chat log"*).

To bridge this fundamental gap, we introduce a synergistic solution comprising a large-scale, fine-grained dataset and an effective frequency-enhanced lightweight mechanism. First, we present VPD-100K, a Visual Privacy Dataset to directly address the aforementioned data challenges. We establish a holistic taxonomy comprising four primary domains: Human Presence, On-Screen PII, Physical Identifiers, and Location Indicators. Through multi-source aggregation and ethical scenario reconstruction, we present 100,000 high-resolution images, where over half exceed 1080*p* to reflect real-world resolution requirements. This large-scale benchmark comprises 33 fine-grained classes and upwards of 190,000 object instances. Statistical analysis reveals that VPD-100K features long-tailed distributions, small object scales, and high visual complexity, providing a rich and diverse foundation for training the next generation of privacy-aware models.

Building upon VPD-100K, we further advance the detection paradigm by proposing an effective and lightweight frequency-enhanced mechanism. It comprises two synergistic modules, *i).* Frequency-Domain Attention Fusion to better perceive the subtle object in frequency spectral, *ii).* Adaptive Spectral Gating Mechanism to learn adaptive gating operations for different high-frequency bands, and *iii).* Frequency-Consistency Loss to enforce spectral alignment by penalizing the feature-level discrepancies in the frequency domain. Spatial-based detectors often struggle with "camouflaged" or tiny sensitive content, such as verification codes that occupy less than 10% of the image area. Our module breaks the limitations of spatial pixel intensity by operating in the frequency domain, remapping features to better capture the subtle high-frequency details of sensitive textual and structural information. This ensures that the model maintains high performance even under the extreme scale variations and low-contrast conditions typical of live streaming scenarios in a lightweight manner. We comprehensively evaluate our framework on both diverse image and streaming video benchmarks, demonstrating that VPD-100K significantly enhances the generalization and robustness of privacy detection across unconstrained environments. Our main contributions are:

- We introduce VPD-100K, a large-scale, fine-grained dataset for visual privacy detection, addressing the critical data gaps in scale, taxonomy, and domain coverage.
- We propose a lightweight Frequency-Enhanced Mechanism, integrating Frequency-Domain Attention Fusion for subtle cue amplification, Adaptive Spectral Gating for dynamic frequency-band calibration, and Frequency-

Consistency loss to minimize cross-spectral distances.

- We perform a comprehensive user study to bridge the gap between VPD-100K detection and subjective privacy perception. The results robustly validate our holistic taxonomy, and also demonstrate that our VPD-100K and methodology effectively align with human experts' judgment in identifying high-risk privacy leaks in unconstrained environments.

## 2. Proposed Dataset

Existing privacy-oriented datasets have laid the groundwork for understanding sensitive information in visual media, yet they exhibit significant limitations that hinder the development of robust, generalized detection models. As summarized in Table 1, prior works often suffer from limited scale or restricted availability. More critically, they lack comprehensive domain coverage due to narrow collection sources. To address these gaps, we construct a large-scale, fine-grained dataset explicitly designed to cover the full spectrum of privacy risks in complex unconstrained streaming environments.

### 2.1. Taxonomy and Data Collection

Driven by the goal of overcoming the domain narrowness of previous datasets (see Table 2), we adopt a taxonomy-driven collection strategy. We first establish a comprehensive taxonomy comprising four primary domains: *Human Presence*, *On-Screen PII*, *Physical Identifiers*, and *Location Indicators*. To ensure each domain is populated with diverse and challenging samples, we implement a targeted multi-source aggregation pipeline.

● *Human Presence.* Privacy protection regulations necessitate distinguishing faces by age and environmental context. However, existing datasets like BIV-Priv (Sharma et al., 2023) exclude human subjects, while PrivacyAlert (Zhao et al., 2022) provides only coarse image-level tags (e.g., "other people"). To bridge this gap, we integrate a subset of WIDER FACE (Yang et al., 2016) and significantly expand the domain by extracting snapshots from online videos. We annotate specific instances with fine-grained attributes (e.g., `child face indoor`), ensuring the dataset captures the high visual complexity and unconstrained poses typical of real-world streaming.

● *On-Screen PII.* This domain presents the most significant challenge and is widely ignored by existing datasets due to ethical constraints. Faced with the impossibility of legally collecting real user data, we employ an ethical scenario reconstruction strategy. Our research team uses internal accounts to simulate realistic digital interactions—such as logging into banking portals or receiving verification codes—and capture high-fidelity screenshots. This process

*Table 1.* Comparison with existing privacy-focused datasets. Our benchmark is constructed from public photos and frames extracted from online video streams, expanding PII coverage beyond image-only sources. **CV ↓**: the Coefficient of Variation of class distribution.

| Dataset | Data Source | Scale | # Classes | # Obj. / Img. | Top-20% Conc. ↓ | CV ↓ | Availability |
|---|---|---|---|---|---|---|---|
| PrivacyAlert (Zhao et al., 2022) | Public Photos from Flickr | 6.8k | 10 | N/A | N/A | N/A | ✕ (Link Inactive) |
| BIV-Priv(Sharma et al., 2023) | Props Shot by Visually Impaired Users | 0.7k | 14 | ~1.0 | By Design | By Design | ✕ (Not Released) |
| DIPA (Xu et al., 2023) | Filtered from OpenImage & LVIS | 1.5k | 25 | ~1.7 | 71.0% | 2.35 | ✕ (Link Inactive) |
| DIPA2 (Xu et al., 2024) | Re-annotated from DIPA | 1.3k | 22 | ~1.4 | 79.0% | 2.50 | ✓ |
| SensitivAlert (Kqiku & Reinhardt, 2024) | Re-labeled from PicAlert/PrivacyAlert | 5.7k | 10 | N/A | N/A | N/A | ✕ (Not Released) |
| BIV-Priv-Seg (Tseng et al., 2025) | Re-annotated from BIV-Priv | 1k | 16 | ~0.94 | By Design | By Design | Eval. Server Only |
| **VPD-100K (Ours)** | **Public Web Photos & Video Streams** | **100k** | **30** | **~1.9** | **62%** | **1.47** | **✓** |

*Table 2.* Granularity and category coverage of privacy datasets across privacy domains. Symbols indicate annotation granularity (● fine-grained, multi-class; ◑ object annotations; ○ whole image annotations; – unsupported). Numbers in parentheses indicate the number of distinct privacy categories per domain, with superscript [↑] marking the maximum value. The complete list of VPD-100K categories is provided in Appendix E.

| Dataset | Human | On-Screen PII | Physical ID | Location |
|---|---|---|---|---|
| PrivacyAlert | ○ (1) | – | ○ (1) | – |
| BIV-Priv | – | – | ◑ (8) | ◑ (1) |
| DIPA | ◑ (2) | ◑ (1) | ◑ (2) | ◑ (2) |
| DIPA2 | ◑ (2) | ◑ (1) | ◑ (2) | ◑ (2) |
| SensitivAlert | ○ (1) | – | ○ (1) | – |
| BIV-Priv-Seg | – | – | ◑ (8) | ◑ (1) |
| **VPD-100K (Ours)** | **● (8[↑])** | **● (9[↑])** | **● (12[↑])** | **● (4[↑])** |

yields pixel-accurate representations of sensitive interfaces (e.g., `chat`, `password`) without infringing on any individual's privacy.

● *Physical Identifiers.* This domain covers tangible items that reveal identity. While BIV-Priv (Sharma et al., 2023) focuses heavily on props (e.g., pill bottles, home letters) and DIPA (Xu et al., 2023) targets general categories, they often overlook common daily-life identifiers found in courier and travel scenarios. We utilize specialized datasets like MIDV-500 (Arlazarov et al., 2019) and perform targeted crawling for under-represented objects such as `train ticket`, `express order`, and `receipt`.

● *Location Indicators.* Detecting location leakage requires a wide variety of environmental text markers (Chaaya et al., 2019). To support this task, we extract snapshots from outdoor shoots, capturing objects such as `street sign`, `store sign`, and `community sign`. These samples exhibit realistic variations in viewing angle and occlusion—conditions often simplified in idealized street-view imagery (Aggarwal & Chauhan, 2025).

Through this targeted collection, we construct a large-scale corpus containing 100,000 images with 33 fine-grained classes, significantly exceeding the scale and category breadth of prior privacy-oriented datasets.

### 2.2. Professional Annotation

To ensure data reliability for model training, we implement a rigorous human-in-the-loop protocol. This semi-automated pipeline integrates expert verification to handle

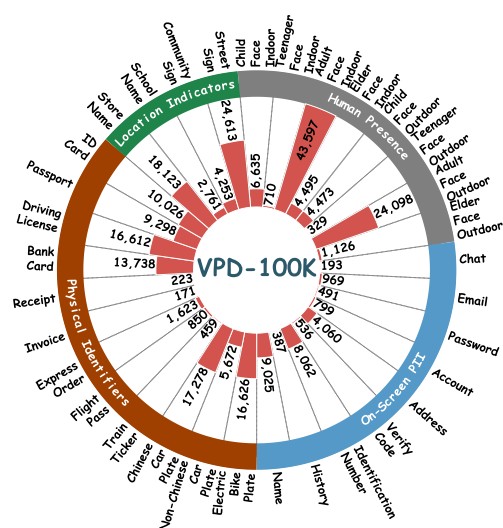

*Figure 1.* The overview of our taxonomy.

our fine-grained privacy taxonomy and large-scale data.

● *Semi-Automatic Pipeline & Challenges.* To efficiently handle the massive scale of 100k images, we employ a hybrid pipeline incorporating object detection and OCR to generate initial predictions (Monteiro et al., 2023). However, this process reveals that automated models frequently struggle with the fine-grained nature of privacy attributes. Specifically, extremely small or dense text fields (*e.g.,* `verify codes`, `account numbers`) are often missed or inaccurately localized due to their limited pixel footprint. This necessitates a labor-intensive manual process where annotators recover missed instances to improve recall and refine the predicted boundaries to ensure the boxes tightly enclose the sensitive content, minimizing background noise (Monarch, 2021; Nadj et al., 2020).

● *Outcome and Quality Control.* We establish strict quality control guidelines to handle visual ambiguities. The final dataset provides precise bounding boxes and class labels for over 190,000 objects. The combination of model-assisted pre-annotation and expert refinement ensures that our dataset offers a challenging and reliable testbed for privacy detection tasks.

### 2.3. Dataset Features and Statistics

We provide a statistical analysis of the primary characteristics of the dataset. These metrics highlight the diversity and

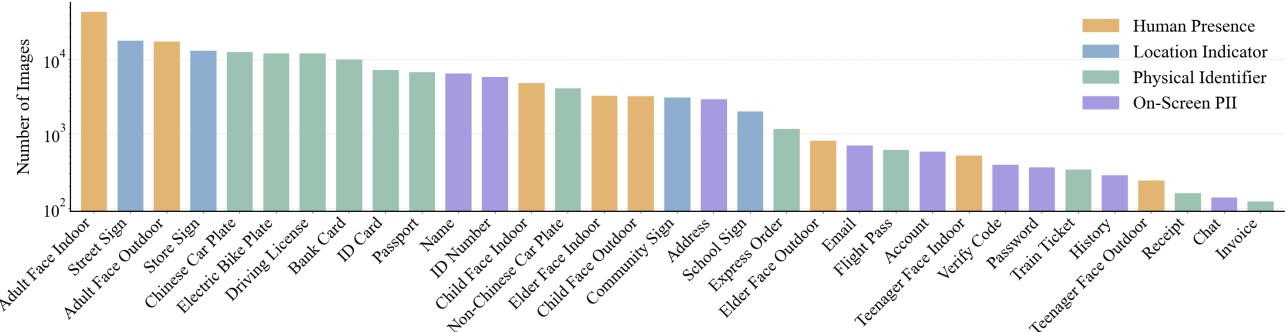

*Figure 2.* Class frequency distribution sorted by frequency. A square root scale is applied to ensure visual readability, accounting for the inherent long-tail characteristic of such datasets.

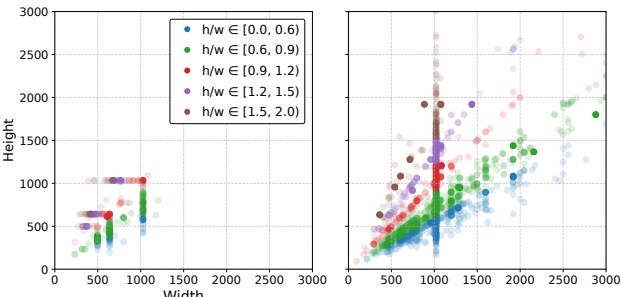

*Figure 3.* Resolution scatter plots of DIPA2 (left) and our dataset (right). Each point corresponds to a single image, plotted by its width and height, and colored by aspect ratio ($h/w$). Compared to DIPA2, our dataset contains substantially more samples, exhibits noticeably higher overall resolution, and spans a wider range of aspect ratios.

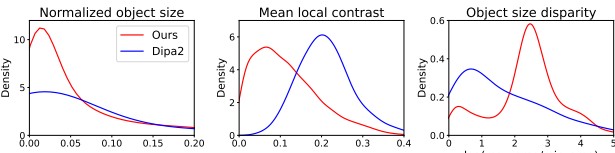

*Figure 4.* Distributions of three object-level statistics for our dataset (red) and DIPA2 (blue). **Left:** Normalized object size, defined as the ratio between each bounding box area and the corresponding image area. **Middle:** Relative object contrast, defined as the ratio of bounding box variance to global scene variance. **Right:** Object size disparity, defined as $\log(\mathrm{max\_area}/\mathrm{min\_area})$ for images with at least two objects.

complexity of the collected images, confirming their value as a challenging testbed for privacy detection models.

● *Class Frequency Distribution.* Figure 2 illustrates the distribution of instances across all classes. It exhibits a significant long-tailed characteristic. While faces and common identifiers appear frequently, sensitive categories like `passport` are naturally scarcer. This requires models to possess strong robustness against class imbalance, reflecting the real-world probability of privacy leakage.

● *Resolution and Aspect Ratio.* Figure 3 compares the image resolution and aspect ratio distributions. Over half of our images exceed 1080p, providing necessary details for small text recognition. Notably, the dataset spans a sub-

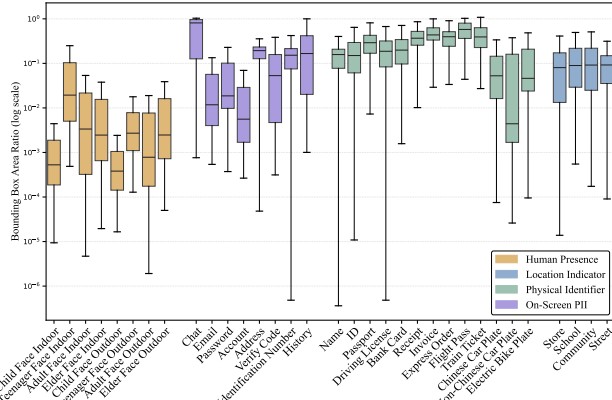

*Figure 5.* Distributions of relative object scales for our dataset. A notable portion of objects appear at small relative scales.

stantially wider range of aspect ratios than the comparison dataset. This diversity prevents models from overfitting to fixed geometric viewports.

● *Normalized Object Size.* Figure 4 (left) compares the normalized object size distributions. Our dataset shows a significantly higher proportion of small objects (occupying < 10% of area) compared to DIPA2. This imposes a stricter requirement for detectors to maintain high recall on tiny targets, such as `verify codes` or distant `ID card`.

● *Relative Object Contrast.* To evaluate visual saliency, we compute the relative contrast, defined as the intensity variance of the target object normalized by the global scene variance (Li et al., 2014). The statistics in Figure 4 (middle) show that our objects tend to have lower contrast ratios. This implies that privacy instances are often "camouflaged" within complex backgrounds, making them significantly harder to distinguish than typical dataset objects.

● *Object Size Disparity.* Figure 4 (right) reports the size disparity within single images. Our dataset exhibits a broader tail, meaning multiple objects with drastically different scales (e.g., a foreground face vs. a background receipt) frequently coexist. This strong multi-scale variation challenges models to handle extreme scale differences within a single forward pass.

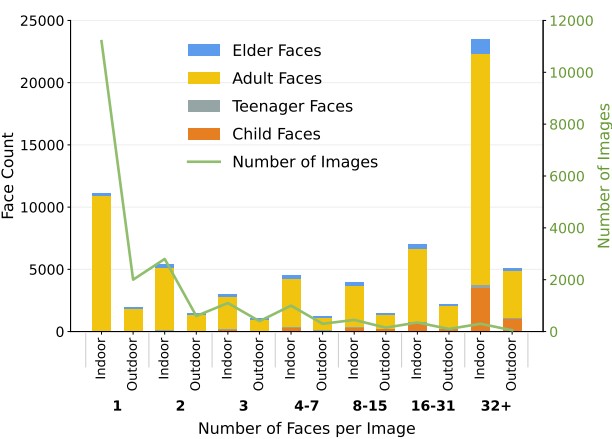

*Figure 6.* Instance count of faces by age group and environment. The line plot shows the total number of images (right axis) for each category.

• *Relative Scale Variance.* Figure 5 presents the relative object size range for each category. The large variance within individual classes (e.g., ID card) confirms that objects appear at diverse distances—from surveillance-like far views to close-up interactions. This wide distribution effectively prevents models from relying on naive size priors and encourages the learning of scale-invariant features.

• *Face Density and Attributes.* We analyze the Human Presence domain in Figure 6 to evaluate crowd density challenges. The green line (image count) follows a long-tail distribution, indicating a solid foundation of sparse scenes (1 and 2 faces) typical of vlogs. Crucially, the bar chart reveals a massive number of face instances in the "32+" category. This indicates that although images with extreme crowds are fewer, they contain a vast amount of targets. This unique distribution ensures the dataset tests model performance across the full spectrum.

## 3. Methodology

### 3.1. Overview of the Frequency-Enhanced Mechanism

While YOLO models demonstrate exceptional performance in general object detection (Redmon et al., 2016; Ali & Zhang, 2024), privacy detection tasks, such as identifying screen tiny text or blurred faces, exhibit a high dependency on high-frequency texture information (Geirhos et al., 2018; Zhang et al., 2025). To address this, we extend the YOLO feature extraction from a purely "spatial domain" approach to a "spatial-frequency dual-stream" architecture. As illustrated in Figure 7, our frequency-enhanced mechanism consists of three synergistic components: *i).* Frequency-Domain Attention Fusion (FDAF) module, embedded within the deep feature aggregation to capture global frequency spectral dependencies; *ii).* Adaptive Spectral Gating Mechanism, designed to implement band-relevant modulation for various high-frequency spec-

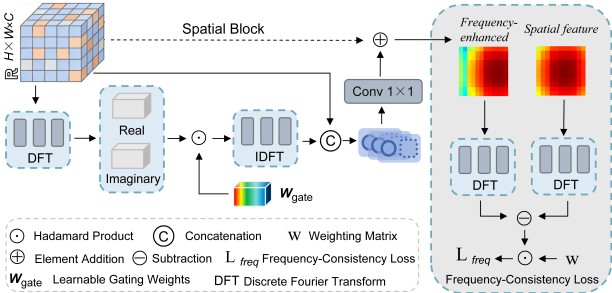

*Figure 7.* The YOLOv10 framework incorporating the frequency domain module within the Neck architecture.

tra. *iii).* Frequency-Consistency Loss to minimize the feature distance in the frequency domain.

### 3.2. Frequency-Domain Attention Fusion Module

To explicitly enhance high-frequency signals during feature pyramid fusion process, we introduce the Frequency-Domain Attention Fusion (FDAF) module into the high-level semantic feature maps of the YOLOv10 neck. This module comprises two sub-processes: Fourier Spectral Transformation, and Cross-Domain Feature Fusion.

**Fourier Spectral Transformation.** Given an input feature $X \in \mathbb{R}^{C \times H \times W}$, we first project it from the spatial domain to the frequency domain. To capture global contextual information, we apply discrete fourier transform (DFT) to each channel independently. Let $X_c(h, w)$ represent the value of the $c$-th channel at spatial coordinates $(h, w)$. Its frequency domain representation $F_c(u, v)$ is calculated as:

$$F_c(u, v) = \sum_{h=0}^{H-1} \sum_{w=0}^{W-1} X_c(h, w) e^{-j2\pi(\frac{uh}{H} + \frac{vw}{W})} \quad (1)$$

where $u, v$ are the frequency coordinates. The resulting $F \in \mathbb{C}^{C \times H \times W}$ contains the amplitude and phase spectra of the layer's features. The amplitude spectrum reflects the texture intensity of the image, while the phase spectrum encodes structural position information.

**Cross-Domain Feature Fusion.** The modulated spectrum $\tilde{F}$ is achieved by the Adaptive Spectral Gating on the $F$ (Sec. 3.3) and then restored to the spatial domain feature $Y_{spatial}$ via the Inverse Discrete Fourier Transform (IDFT):

$$Y_{spa} = \mathcal{R} \left( \frac{1}{HW} \sum_{u=0}^{H-1} \sum_{v=0}^{W-1} \tilde{F}_c(u, v) e^{j2\pi(\frac{uh}{H} + \frac{vw}{W})} \right) \quad (2)$$

where $\mathcal{R}(\cdot)$ denotes the operation of taking the real part. Finally, to compensate for potential local spatial information loss caused by spectral operations, we employ a residual connection structure. The original input $I$ is fused with

the frequency-enhanced feature $Y_{spa}$, followed by channel integration via a $1 \times 1$ convolution layer:

$$I_{out} = \text{Conv}_{1 \times 1}(\text{Concat}(I, Y_{spa})) + I \quad (3)$$

### 3.3. Adaptive Spectral Gating Mechanism

The learnable spectral gating weights $W_{gate}$ act as an adaptive band-pass filter. It can automatically adjust the band according to the structural characteristics of the privacy object. For instance, for text-type privacy objects, the spectral mask tends to exhibit strong activation in horizontal and vertical frequency components, reflecting the stroke characteristics of text. This mechanism allows the model to focus on specific frequency enhancements rather than blindly amplifying all high-frequency signals. To adaptively screen for key frequencies (typically high-frequency parts representing details), we design a learnable gating operator in the complex domain. We define a learnable weight tensor $W_{gate} \in \mathbb{R}^{C \times H \times W}$ and modulate the spectrum via the Hadamard product:

$$\tilde{F}_c(u, v) = F_c(u, v) \odot \sigma(W_{gate}(u, v)) \quad (4)$$

where $\sigma(\cdot)$ is the Sigmoid activation function, used to normalize weights to the $(0, 1)$ interval, acting as a "soft mask". This mechanism allows the network to automatically suppress background noise and enhance the feature response of privacy objects.

### 3.4. Frequency-Consistency Loss

To supervise the learning of the FDAF module, in addition to the standard YOLO losses ($\mathcal{L}_{box}, \mathcal{L}_{cls}, \mathcal{L}_{dfl}$), we introduce a Frequency-Consistency Loss, $\mathcal{L}_{freq}$. This loss aims to minimize the distance between the predicted box region features and the Ground Truth in the frequency domain.

We define $\mathcal{L}_{freq}$ as the weighted Euclidean distance between the predicted feature $P$ and the target feature $T$ in the frequency domain:

$$\mathcal{L}_{freq} = \frac{1}{N} \sum_{i=1}^{N} ||W \odot (\mathcal{F}(P_i) - \mathcal{F}(T_i))||_2^2 \quad (5)$$

where $P_i$ and $T_i$ are the feature maps within the $i$-th detection box, and $N$ is the number of targets in the batch. $W$ is a static frequency weighting matrix, where the value increases with frequency $r$, specifically $w(r) = 1 + \lambda \cdot r$. This design forces the model to prioritize fitting the high-frequency boundary information of privacy objects during fine-tuning, thereby improving detection precision.

The total loss function is defined as:

$$\mathcal{L}_{total} = \mathcal{L}_{yolo} + \beta \cdot \mathcal{L}_{freq} \quad (6)$$

where $\beta$ is a balancing hyperparameter. The balancing hyperparameter $\beta$ is set to 0.05 to maintain a stable trade-off between the standard detection loss and the frequency-consistency loss. Given that the feature maps in the Neck module are highly sensitive in the frequency domain, a relatively small $\beta$ prevents the high-frequency gradients from overwhelming the optimization process, ensuring that the model refines boundary details without compromising the convergence of the primary detection task.

## 4. Benchmark Experiments

Implementation details and baselines information can be found in the Appendix D.

### 4.1. Results and Data Analysis

**Performance on image data.** From Table 3, our proposed framework achieves superior performance across the majority of metrics on image data, significantly outperforming the other 14 baselines. Specifically, our model achieves the highest AP scores with 58.6% for $AP^{val}$ and 73.4% for $AP_{50}$, representing substantial improvements of 8.9% and 4.7% respectively, over the second-best performance. For small objects ($AP_{small}^{val}$), our approach achieves 36.5%, demonstrating robust detection capabilities even at challenging scales. Our model also achieves the best performance for medium and large objects with AP of 62.3% and 70.6%, while maintaining an impressive F1-Score of 0.81. Some visual examples of our method are in Figure 8.

**Performance on live streaming video.** For streaming video, our framework demonstrates strong real-time detection capabilities with competitive accuracy, as shown in Table 4. The model achieves a latency of only 7.51$ms$, enabling smooth processing at over 130 FPS, which is key for live-streaming applications. This represents a significant efficiency improvement compared to resource-intensive baselines like Grounding-DINO (119.5$ms$). While the streaming context presents additional challenges compared to image detection, which is reflected in slightly lower AP scores, our model maintains robust performance with the highest F1-Score of 0.81 among all baselines. Notably, our approach balances the trade-off between accuracy and inference speed more effectively than lightweight alternatives, such as the fastest Yolov10s model with 2.53$ms$ latency but only 0.65 F1-Score.

**Performance on real-world out-of-distribution data.** To assess the generalizability and applicability of our dataset in practical scenarios, we conduct a qualitative evaluation on real-world videos collected from live-streaming plat-

*Table 3.* Quantitative results of different baseline approaches on VPD-100K image test dataset. The best scores are highlighted in **bold**. The training-based baselines are fine-tuned by our VPD-100K training set to enable the capability of the generalizable privacy protection. For brevity, we denote our Frequency-Enhanced Mechanism as FEM.

| Baselines | $AP^V$ | $AP_{50}^V$ | $AP_{75}^V$ | $AP_S^{val}$ | $AP_M^V$ | $AP_L^V$ | GFLOPs | Latency (ms) | F1-Score |
|---|---|---|---|---|---|---|---|---|---|
| Grounding-DINO | 48.1 | 65.8 | **62.6** | 30.4 | 51.3 | 62.3 | 464.0 | 119.5 | 0.68 |
| FBRT-YOLO | 20.2 | 45.8 | 42.2 | 28.1 | 46.3 | 56.2 | 22.9 | 3.72 | 0.43 |
| DEIM-D-FINE-S | 49.0 | 65.9 | 53.1 | 30.4 | 52.6 | 65.7 | 26.0 | 3.46 | 0.69 |
| Gold-YOLO-S | 46.4 | 63.4 | 52.2 | 25.3 | 51.3 | 63.6 | 46.0 | 3.82 | 0.66 |
| Gold-YOLO-L | 52.7 | 70.1 | 58.0 | 32.1 | 57.0 | 70.1 | 153.8 | 10.91 | 0.74 |
| YOLOv7-tiny | 38.3 | 46.6 | 45.3 | 19.1 | 44.2 | 54.1 | **12.6** | 5.43 | 0.55 |
| YOLOv7 | 51.1 | 50.1 | 59.1 | 32.6 | 58.1 | 68.0 | 99.7 | 7.12 | 0.71 |
| YOLOv8s | 44.3 | 60.5 | 51.3 | 24.1 | 50.8 | 59.8 | 26.0 | 7.13 | 0.64 |
| YOLOv8L | 52.6 | 68.3 | 59.1 | 32.6 | 58.5 | 67.3 | 152.0 | 14.76 | 0.72 |
| YOLOv9s | 46.0 | 62.3 | 50.8 | 25.6 | 53.0 | 62.5 | 24.0 | 2.73 | 0.67 |
| YOLOv9L | 53.4 | 68.6 | 57.9 | 33.9 | 59.1 | 70.3 | 124.0 | 7.73 | 0.73 |
| YOLOv10s | 46.3 | 62.7 | 51.3 | 26.1 | 53.2 | 62.7 | 23.0 | **2.53** | 0.65 |
| YOLOv10L | 53.8 | 69.6 | 58.4 | 33.6 | 59.8 | 70.8 | 121.0 | 7.42 | 0.73 |
| **YOLOv10s+our FEM** | 52.1 | 67.1 | 54.6 | 30.1 | 55.6 | 64.3 | 26.0 | 2.71 | 0.71 |
| **YOLOv10L+our FEM** | **58.6** | **73.4** | 61.3 | **36.5** | **62.3** | **70.6** | 132.0 | 7.51 | **0.81** |

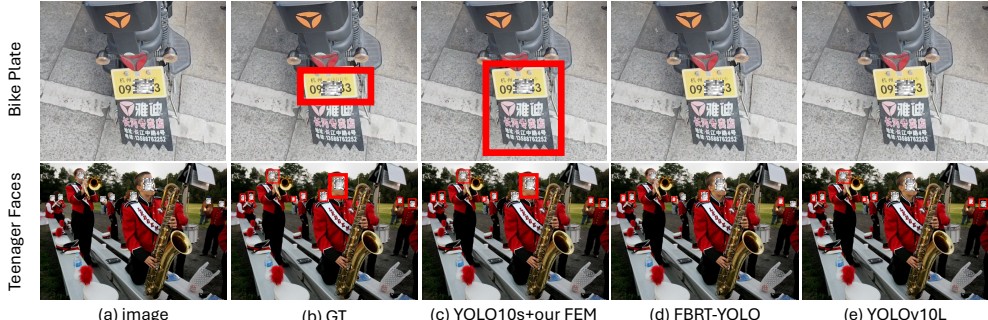

*Figure 8.* Visual performance of the proposed Frequency-Enhanced Mechanism.

*Table 4.* Quantitative results of different real-time baseline approaches on VPD-100K live streaming video dataset. The best scores are highlighted in **bold**. The all baselines are fine-tuned on our VPD-100K training set to enable the capability of the generalizable privacy protection. For brevity, we denote our Frequency-Enhanced Mechanism as FEM.

| Baselines | $AP^V$ | $AP_{50}^V$ | $AP_{75}^V$ | $AP_S^V$ | $AP_M^V$ | $AP_L^V$ |
|---|---|---|---|---|---|---|
| FBRT-YOLO | 20.2 | 45.8 | 42.2 | 28.1 | 46.3 | 56.2 |
| DEIM-D-FINE-S | 49.1 | 66.1 | 53.5 | 30.5 | 52.8 | 65.9 |
| Gold-YOLO-S | 46.4 | 63.4 | 52.2 | 25.3 | 51.3 | 63.6 |
| Gold-YOLO-L | 52.5 | 69.7 | 57.7 | 31.8 | 56.9 | 69.8 |
| Yolov7-tiny | 38.0 | 45.7 | 45.3 | 18.7 | 43.8 | 53.9 |
| Yolov7 | 50.9 | 49.7 | 58.8 | 31.4 | 57.6 | 67.8 |
| Yolov8s | 44.3 | 60.5 | 51.3 | 24.1 | 50.8 | 59.8 |
| Yolov8L | 52.5 | 67.5 | 58.7 | 32.0 | 58.5 | 67.3 |
| Yolov9s | 46.0 | 62.3 | 50.8 | 25.6 | 53.0 | 62.5 |
| Yolov9L | 53.4 | 68.6 | 57.9 | 32.8 | 59.1 | 70.3 |
| Yolov10s | 45.6 | 61.9 | 49.9 | 25.4 | 52.4 | 61.9 |
| Yolov10L | 52.9 | 68.2 | 57.6 | 32.8 | 59.1 | 69.7 |
| **YOLO10s+our FEM** | 51.8 | 66.4 | 53.9 | 28.9 | 55.2 | 63.8 |
| **YOLO10L+our FEM** | **57.7** | **72.8** | **60.4** | **36.1** | **61.9** | 70.0 |

*Table 5.* Ablation study of Frequency-enhanced Mechanism on YOLOv10-S basemodel.

| Model | FDAF | LSG (Gating) | $\mathcal{L}_{freq}$ (Loss) | AP | $AP_{50}$ | $AP_{75}$ | $AP_S$ |
|---|---|---|---|---|---|---|---|
| I (Base) | - | - | - | 46.3 | 62.7 | 51.3 | 26.1 |
| II | ✓ | - | - | 48.5 | 64.2 | 52.8 | 27.5 |
| III | ✓ | ✓ | - | 50.9 | 65.8 | 53.9 | 29.2 |
| **IV (Ours)** | ✓ | ✓ | ✓ | **52.1** | **67.1** | **54.6** | **30.1** |

### 4.2. Ablation Study

Using YOLOv10-S as the baseline model, we sequentially introduce the Frequency-Domain Attention Fusion (FDAF) structure, Learnable Spectral Gating (LSG), and Frequency-Consistency Loss ($\mathcal{L}_{freq}$). The results are shown in Table 5.

**Effectiveness of Frequency Domain Information.** As shown in Model II, simply introducing a DFT-based frequency branch in the Neck (without gating, performing only feature fusion) improved AP from 46.3% to 48.5%. This significant improvement demonstrates the importance of expanding the feature perspective from the pure "spatial domain" to a "space-frequency dual-stream".

**Impact of Learnable Spectral Gating.** In Model III, we add the LSG module to the frequency branch. The results show that AP increases further by 2.4%, with a notable improvement in $AP_S$ (small objects) (from 27.5% to 29.2%), which validates the hypothesis: not all frequency components are beneficial for privacy detection. LSG

forms. In contrast to our self-constructed dataset, these videos exhibit previously unseen domain shifts, including unstable handheld camera motion and complex screen-sharing scenarios. As shown in Figure 9, our framework, trained exclusively on our dataset, successfully detects multiple forms of privacy exposure in the wild. These include sensitive personal information on the receipt and unintended information leakage during screen sharing. These qualitative results demonstrate that models trained on our dataset can generalize beyond the training domain and remain effective in real-world live-streaming scenarios.

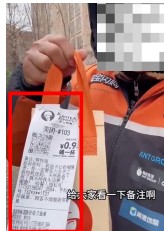 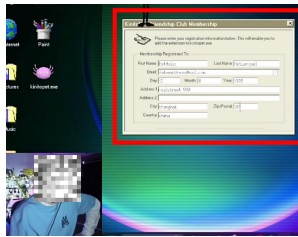

*(a)* Receipt with personal data    *(b)* Screen-sharing privacy

*Figure 9.* Privacy exposure in real-world live streaming scenarios identified by our tool.

acts as an adaptive filter, suppressing high-frequency background noise interference while enhancing specific texture frequencies required for privacy objects (*e.g.,* text, faces), thereby improving feature purity.

**Contribution of Frequency-Consistency Loss.** Model IV demonstrates the performance of the full method. After introducing $\mathcal{L}_{freq}$, the model gains an additional 1.2% in AP and achieves the best performance in the high-precision metric $AP_{75}$ (54.6%). This indicates that structural fusion alone is insufficient. It is necessary to explicitly minimize the distance between the predictor and the target features in the frequency domain through $\mathcal{L}_{freq}$. This loss function serves as a powerful regularizer,forcing the network to prioritize fitting high-frequency boundary information during fine-tuning, thereby improving the localization quality of detection boxes.

## 5. Usability Evaluation

We perform a light-weight usability evaluation to demonstrate completeness of the proposed privacy instance taxonomy and the perceived effectiveness of our privacy instance detection framework. The study protocol was reviewed and approved by the university's Ethical Review Board.

**Experimental Settings.** Twenty Participants first complete a brief questionnaire collecting demographic information and their live streaming experience. Participants are then shown a short system introduction video (3-4 minutes), which provides a non-technical overview of the system's goals, illustrative examples of privacy instances appearing in live streams, and an explanation of the proposed privacy instance taxonomy. The study consists of two main parts. In Part 1, participants evaluate the proposed privacy instance taxonomy specifically. In Part 2, participants evaluate the perceived usefulness, trustworthiness, and usability of the proposed framework by rating eight statements on a 5-point Likert scale. The complete study protocol is provided in the appendix.

**Results.** The pie chart of Figure 10 summarizes participants' assessments of the completeness of the proposed privacy instance taxonomy. The pie charts show the distribution of responses across the four top-level categories and

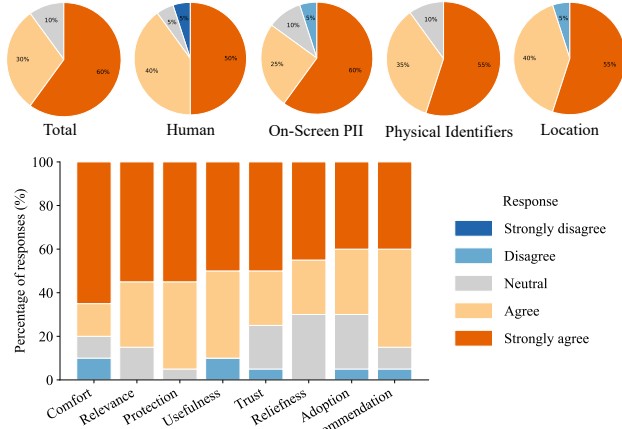

*Figure 10.* Distribution of Likert-scale ratings across evaluation dimensions. The top pie charts are the completeness of privacy instance taxonomy, and the bottom stacked bar charts are the rating of our proposed solution.

each sub-category. A clear majority of participants select Agree or Strongly agree, with the overall assessment reaching 90% positive responses. This indicates that participants generally perceive the taxonomy as complete and representative of privacy-sensitive instances that may appear during live streaming. Figure 10 (stacked bar charts) presents participants' perceptions of the proposed framework across eight usability and perception dimensions measured using 5-point Likert-scale questions. Overall, responses skew strongly positive, with Agree and Strongly agree dominating all dimensions. Especially, the majority of participants indicate that the system is useful for real-time live streaming scenarios (**Usefulness**). Also, most participants report that using the system would make them feel more comfortable while live streaming (**Comfort**), suggesting perceived value in reducing privacy-related anxiety. Moreover, responses to show that a substantial proportion of participants would consider using such a system in their own live streams, indicating promising acceptance and practical relevance (**Adoption**).

## 6. Conclusion

In this paper, we address the critical bottleneck in visual privacy protection by introducing VPD-100K, a large-scale, fine-grained dataset that reflects the complexity of real-world sensitive information. By establishing a frequency-enhanced lightweight detection module, we enhance the perception of subtle high-frequency details crucial for identifying sensitive structural and textual information. Extensive benchmarks on both static images and dynamic streaming videos validate the robustness and generalizability of our approach. Beyond its performance gains, this work provides a foundational benchmark for the community, paving the way for more resilient and privacy-aware intelligent systems in the increasingly transparent digital era.

## Impact Statement

This research introduces VPD-100K, a large-scale dataset designed to enhance visual privacy protection. Our research carries significant positive social implications by providing robust technical solutions to mitigate unintentional privacy leakage in unconstrained environments, such as live streaming and digital media sharing. We emphasize that the primary objective of this study is defensive: to empower users and platforms with tools that accurately identify and redact sensitive information. Crucially, the study protocol was reviewed and approved by the university's Ethical Review Board (ERB). We advocate for the responsible use of this dataset in strict accordance with legal frameworks.

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

# Appendix

## A. Ethical Considerations

This study was approved by the University's Human Research Ethics Committee. Given that our research focuses on privacy protection mechanisms in live-streaming scenarios, particular care was taken to safeguard participants' privacy, autonomy, and well-being throughout the study.

All participants provided informed consent prior to participation. Participants were informed of the study's purpose, the types of questions involved, and their right to withdraw at any time without penalty. To minimize potential discomfort when reflecting on privacy-related incidents, participants were clearly advised that they could skip any questions they found sensitive or uncomfortable.

As the study concerns visual privacy risks in live streaming, we deliberately avoided collecting identifiable personal data, raw live-stream footage, or screenshots containing sensitive visual information. Survey questions were designed to focus on participants' experiences and perceptions rather than requiring them to share or reproduce privacy-invasive content.

All collected data were anonymized at the point of collection and stored securely on encrypted systems accessible only to the research team. Any potentially identifying information was removed during transcription and analysis. Particular attention was paid to preventing re-identification risks arising from contextual descriptions of live-streaming scenarios.

Overall, the study was designed to minimize privacy risks while enabling participants to reflect on and discuss privacy protection practices in live-streaming environments in a safe and controlled manner.

## B. FigureAppendix

## C. Background and Related Work

Privacy is a fundamental requirement in software system design and deployment, and has been extensively studied across legal, social, and technical domains (Liu et al., 2003; Nissenbaum, 2004; GDPR; CCPA). In the context of computer vision applications, the detection of privacy-sensitive instances has emerged as a critical area of research (Jiang et al., 2024; Sun et al., 2023; Liu et al., 2021; Sun et al., 2021b), aiming to identify and mitigate the exposure of personally identifiable or sensitive information in visual data.

### C.1. Privacy Instance Detection Datasets

Conventional object detection datasets, such as COCO (Lin et al., 2014), lack the fine-grained annotations necessary for specific downstream tasks. Thus, people proposed various datasets for specific downstream tasks. To support research on visual privacy protection, several datasets have been proposed, such as PrivacyAlert (Zhao et al., 2022), DIPA (Xu et al., 2023; 2024), and SensitivAlert (Kqiku & Reinhardt, 2024). These datasets have played an important role in enabling supervised learning for privacy-related tasks. However, **there is no widely-accepted, large-scale, and holistic taxonomy of what constitutes privacy-sensitive content** in interactive contexts, especially when visual, textual, and contextual cues are intertwined. Consequently, existing datasets are not compatible when applied to live streaming scenarios.

### C.2. Privacy in Interactive Context

Interactive context, such as live streaming, contains large volum of complex visual cues (Hu et al., 2025; Qiu et al., 2023), introducing novel privacy risks that differ fundamentally from offline or asynchronous media (Jackson, 2017; Wu et al., 2022; 2023). Existing research on live streaming platform privacy (Jackson, 2017; Zhou & Pun, 2020; Wu et al., 2023) has mainly focused on software users (i.e., viewers), while the privacy risks faced by platform content creators (i.e., streamers) have received far less attention (Wu, 2024). Nevertheless, streamers frequently expose their physical environment, on-screen activities, and personal documents in real-time, often without full awareness of the resulting unintentional privacy leakage (Jackson, 2017; Zhou & Pun, 2020; Wu et al., 2022).

Privacy protection in live streaming presents several unique challenges (Jackson, 2017; Li et al., 2018). Live streaming scenarios are highly complex and unconstrained, spanning diverse indoor and outdoor environments, varying camera viewpoints, and rapidly changing scenes. Also, privacy protection must operate under strict real-time constraints, leaving little room for heavy post-processing or human intervention (Zhou & Pun, 2020; Shamshad et al., 2023; Zhang et al., 2023; Zhao

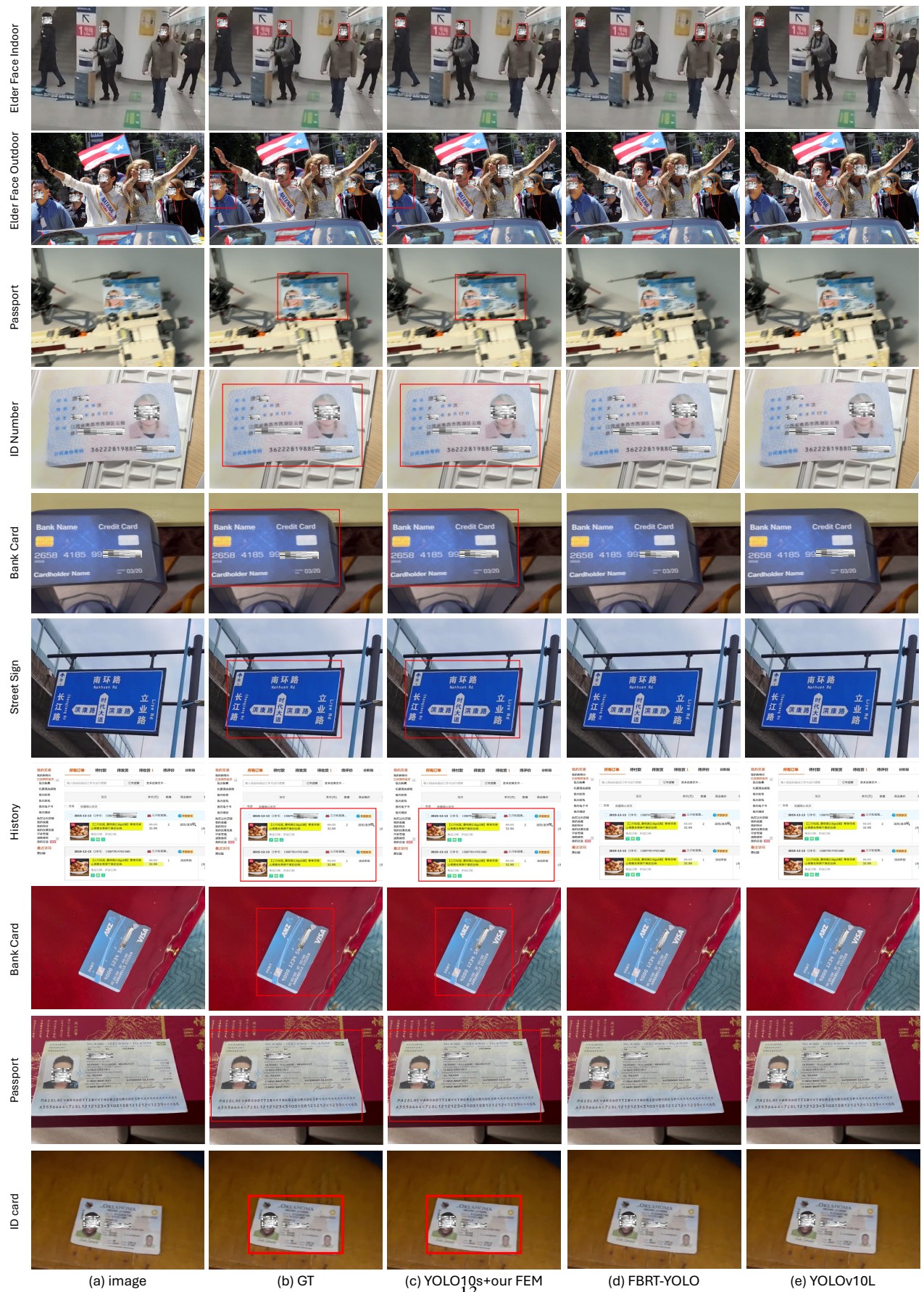

(a) image  (b) GT  (c) YOLO10s+our FEM  (d) FBRT-YOLO  (e) YOLOv10L

*Figure 11.* Visual performance of the proposed Frequency-Enhanced Mechanism.

et al., 2025). These challenges collectively differentiate live streaming privacy from conventional image or video privacy settings.

## D. Implementation Details and Baselines

### D.1. Implementation Details

**Training Settings.** We conduct all experiments in a cluster equipped with NVIDA A100 GPU using the PyTorch framework. Specifically, the total loss function $\mathcal{L}_{total}$ is a weighted sum of the standard YOLO loss and our proposed Frequency-Consistency Loss $\mathcal{L}_{freq}$. To ensure stable convergence, the balance hyperparameter $\beta$ is set to 0.05. The model is trained using the SGD optimizer with momentum set at 0.937 and weight decay set at $5 \times 10^{-4}$.

### D.2. Baselines

To comprehensively evaluate the effectiveness of our Frequency-Enhanced framework, we compare it with a wide range of State-of-the-Art (SOTA) object detectors. The baseline models are categorized as follows:

**Standard Real-time Detectors:** The YOLO series, including YOLOv7 (Tiny/Normal) (Wang et al., 2023b), YOLOv8 (S/L) (Jocher et al., 2023), YOLOv9 (S/L) (Wang et al., 2024b), and the base model YOLOv10 (S/L) (Wang et al., 2024a). These models represent current industrial standards for efficiency and accuracy.

**Advanced and Specialized Detectors:** Gold-YOLO (S/L) (Wang et al., 2023a), known for its information aggregation and distribution mechanism; DEIM-D-FINE-S (Huang et al., 2025), a recent detection Transformer variant; Grounding-DINO (Liu et al., 2024), a representative open-set detector used for benchmarking zero-shot capabilities; and FBRT-YOLO (Xiao et al., 2025), used as a frequency-related baseline for comparison.

## E. Fine-Grained Category Labels

Below is the 33 fine-grained category labels of our dataset, grouped by four core privacy domains (counts in parentheses).

| Domain (Count) | Detailed Labels |
|---|---|
| Human Presence (8) | child face indoor, teenager face indoor, adult face indoor, elder face indoor, child face outdoor, teenager face outdoor, adult face outdoor, elder face outdoor |
| On-Screen PII (9) | chat, email, password, account, address, verify code, identification number, history, name |
| Physical Identifier (12) | id, passport, driving license, bank card, receipt, invoice, express order, flight pass, train ticket, chinese car plate, non-chinese car plate, electric bike plate |
| Location Indicator (4) | store sign, school sign, community sign, street sign |

## F. User Study Questionnaire

**Background Questions**

Q1. What is your age group?
- 18–24
- 25–34
- 35–44
- 45+

Q2. What is your primary role in the live streaming community?
- Streamer (have experience broadcasting content)
- Viewer (primarily watch others' live streams)

**Questions for Streamers**

Q3. How many years of experience do you have with broadcasting live content?

- Less than 1 year
- 1–3 years
- 3–5 years
- More than 5 years

Q4. How frequently do you go live?

- Daily
- Several times a week
- Once a week
- Rarely

Q5. Which devices do you primarily use to host your streams? (Multiple choice)

- Mobile Phone
- Desktop / Laptop
- Tablet / Others

Q6. What are your primary broadcasting categories? (Multiple choice)

- Indoor (Life sharing, Chatting, etc.)
- Outdoor (Vlogging, Travel, Street performance)
- Gaming
- Others

Q7. Have you ever experienced or been concerned about privacy incidents during a live stream?

- Yes, I have had a privacy leak.
- I haven't had one, but I am worried about it.
- No, I am not particularly concerned.

**Questions for Viewers**

Q8 How frequently do you tune into live streams?

- Daily
- Several times a week
- Once a week
- Rarely

Q9 What categories of live streams do you most frequently watch? (Multiple choice)

- Indoor (Life sharing, Chatting, etc.)
- Outdoor (Vlogging, Travel, Street performance)
- Gaming
- Others

Q10 Have you ever spotted privacy-sensitive information being accidentally exposed during a live stream?

- Yes, I have spotted privacy leaks multiple times.
- Yes, I have spotted it once or twice.
- No, I have never noticed any privacy leaks.
- I am not sure / I never paid attention.

**System Introduction**

Participants were shown a short video demonstrating the proposed Privacy Detection System and the detailed subcategories of the Four-Domain Privacy Taxonomy prior to answering the following questions.

Q11 Please rate your level of agreement that if this four-domain framework comprehensively captures privacy-sensitive content in live streaming on a scale of 1 to 5, where 1 represents "Strongly Disagree," and 5 represents "Strongly Agree".

Q12 Are there any additional major privacy dimensions that should be included in the top-level framework?

Q13 Please rate your level of agreement that if the subcategories in Human Presence sufficiently cover the privacy risks

related to human presence on a scale of 1 to 5, where 1 represents "Strongly Disagree," and 5 represents "Strongly Agree".

Q14 Based on your experience, are there any other human-related privacy instances that should be added?

Q15 Are any subcategories within this group unnecessary or unclear?

Q16 Please rate your level of agreement that if the subcategories in On-Screen PII sufficiently cover the potential PII risks on-screen on a scale of 1 to 5, where 1 represents "Strongly Disagree," and 5 represents "Strongly Agree".

Q17 Would you suggest adding any other PII elements that frequently appear during screen sharing?

Q18 Are any subcategories within this group unnecessary or unclear?

Q19 Please rate your level of agreement that if the subcategories in Physical Identifiers sufficiently cover the privacy risks of physical objects on a scale of 1 to 5, where 1 represents "Strongly Disagree," and 5 represents "Strongly Agree".

Q20 Are there any other physical objects or identifiers you suggest adding to this category?

Q21 Are any subcategories within this group unnecessary or unclear?

Q22 Please rate your level of agreement that if the subcategories in Location Indicators is sufficient on a scale of 1 to 5, where 1 represents "Strongly Disagree," and 5 represents "Strongly Agree".

Q23 Are there any other geographic or situational indicators you would recommend including?

Q24 Are there any subcategories within this group unnecessary or unclear?

Q25 Please rate your agreement with the following statements regarding the Privacy Detection System on a scale of 1 to 5, where 1 represents "Strongly Disagree" and 5 represents "Strongly Agree".

- Relevance: Detected privacy instances reflect real streaming scenarios
- Risk Reduction: The system helps reduce accidental privacy disclosure
- Utility: The system is useful in real-time streaming contexts
- Comfort: Using the system increases comfort while streaming
- Balance: The system balances content creation and privacy protection
- Trust: I would trust the system to detect important privacy risks
- Adoption: I would consider using the system myself
- Recommendation: I would recommend the system to others

Q26 Do you have any other comments, suggestions, or concerns regarding the Privacy Detection System or the privacy categories we discussed? We would love to hear your thoughts on how this system could be improved for real-world streaming scenarios.

