# OpenReview forum: "VPD-100K: Towards Generalizable and Fine-grained Visual Privacy Protection"
_ICML.cc/2026/Conference — ICML 2026 regular_

### Official Review · Reviewer_mWVq · 2026-03-10

**Soundness:** 3
**Presentation:** 4
**Significance:** 3
**Originality:** 3
**Overall Recommendation:** 5
**Confidence:** 3

**Summary:**

This paper proposes VPD-100K, a large-scale, fine-grained visual privacy detection dataset designed to address the limitations of existing privacy detection datasets, such as small size, coarse annotation, and narrow domain coverage. The dataset contains 100,000 high-resolution images, covering 33 fine-grained categories and over 190,000 target instances, divided into four core domains: human presence (8 categories), screen PII (9 categories), physical identifiers (12 categories), and location indicators (4 categories). The dataset is characterized by a long-tailed distribution, a high proportion of small targets, and high visual complexity. The authors also propose a frequency-enhanced lightweight detection module, incorporating frequency-domain attention fusion, adaptive spectral gating, and frequency consistency loss to better capture high-frequency details of sensitive information. Experiments demonstrate that this method achieves excellent performance on both image and streaming video benchmarks, significantly improving detection accuracy while maintaining real-time performance.

**Compliance With Llm Reviewing Policy:**

Affirmed.

**Final Justification:**

The author's reply answered my question, and I think revising the paper is acceptable.

**Key Questions For Authors:**

The rigor of the real-time claim: Table 4 claims "130FPS" but fails to specify hardware configuration (only mentioning NVIDIA A100), input resolution, and batch size. Is a 7.51ms latency reasonable for YOLOv10L on the A100? Detailed configuration information is recommended.

**Limitations:**

Yes

**Strengths And Weaknesses:**

Overall Assessment: This is a solid work addressing an important issue in the field of visual privacy protection. The dataset construction and methodological design demonstrate a degree of innovation, and the experiments are comprehensive. However, there is room for improvement in the originality of the methodology, the ethical details of the dataset, and the rigor of some experimental designs.
Strengths:
1. The dataset size (100K images) and fine-grained annotations (33 classes) significantly surpass existing privacy detection datasets (such as DIPA2 with only 1.3K images), filling a gap in the field.
2. The ethical review process is complete (Appendix A), clearly stating that the collection of real user data will be avoided and an "ethical scenario reconstruction" strategy will be adopted.
Weaknesses:
1. Data reproducibility: The paper claims that the dataset will be made public (Availability is marked ✓ in Table 1), but the text mentions "simulating digital interactions using internal accounts." However, it fails to adequately explain the specific generation protocol of this simulated data, whether it contains templated UI elements, and whether it may contain the interface copyrights of real third-party services.
2. The rigor of the real-time claim: Table 4 claims "130FPS" but fails to specify hardware configuration (only mentioning NVIDIA A100), input resolution, and batch size. Is a 7.51ms latency reasonable for YOLOv10L on the A100? Detailed configuration information is recommended.
3. The gain boundary with general detectors is unclear: Table 3 shows a 4.8% improvement in AP compared to the baseline with YOLOv10L+FEM, but is this mainly due to the dataset rather than the method? It is recommended to add a control experiment with "changing only the dataset, not the method".

---

> ### Author Rebuttal · Authors · 2026-03-31
>
> We greatly appreciate your insightful and detailed feedback. Your comments have been carefully discussed and fully taken into account, and we improve our work accordingly.
>
> > **1.Data availability, reproducibility, and the collection protocol**:
>
> We would like to clarify that the 'simulation digital interactions ...' process refers to controlled data collection using researcher-owned accounts in real-world interfaces, rather than fully synthetic or templated UI creation.
>
> Specifically, members of the research team interact with commonly used real-world software (e.g., typing passwords or sending chats to others) using burner accounts without any real personal information, and capture screenshots under realistic usage scenarios.
>
> As for reproducibility, the data collection protocol is fully reproducible. It consists of (i) defining privacy categories, (ii) interacting with the real-world software using burner accounts, and (iii) capturing and annotating screenshots following our taxonomy. We will further clarify this protocol in the revised manuscript to improve transparency.
>
> Regarding the copyright and third-party interface concerns, we take several precautions:
>
> ●All data are collected under a research-only setting with appropriate usage restrictions
>
> ●No personal or sensitive user data is included, and
>
> ●The images are used as part of a broader visual scene for object detection, instead of reproducing or redistributing interface designs as standalone assets.
>
> > **2.Hardware configuration**:
>
> We sincerely thank the reviewer for highlighting the need for these critical hardware configuration details. While standard YOLO benchmarks  [1] [2] typically report latency at a lower 640×640 resolution, our privacy detection task strictly requires a high resolution (1920×1088) to successfully identify extremely small objects, such as tiny faces, verification codes, and on-screen text.
>
> Therefore, the latency reported in Table 4  is measured under one A100 GPU with a larger resolution of 1920×1088 and batch size 1, and without the half-precision quantization. All benchmark baselines are evaluated in the same Hardware configurations for fair comparison.
>
> [1]. YOLOv10: Real-Time End-to-End Object Detection.
>
> [2]. DETRs Beat YOLOs on Real-time Object Detection.
>
> > **3.Finetune clarifications**:
>
> We respectfully make a clarification regarding the experimental setup in Table 3. Because off-the-shelf detectors cannot recognize our novel fine-grained privacy categories, all evaluated baselines, including the vanilla YOLOv10L, are already fine-tuned on our VPD-100K training set to ensure a fair comparison.
>
> VPD-100K provides the essential foundation, enabling general detectors to recognize novel fine-grained privacy categories. Our FEM module explicitly empowers the network to capture subtle high-frequency details, directly yielding the 4.8% performance improvement.
>
> We thank the reviewer for pointing this out, and we will update the manuscript to eliminate this confusion.
>
> **We hope these clarifications and added results address the reviewer’s concerns and improve the overall assessment of the paper.**

---

> > ### Author Rebuttal · Reviewer_mWVq · 2026-04-01
> >
> > I thank the authors for their detailed response and the clarifications provided regarding the dataset and experimental setup.
> > 1: The authors have clarified the data collection protocol, specifying the use of researcher-owned burner accounts. This addresses my concerns regarding privacy and synthetic vs. real-world data. The commitment to transparently documenting this protocol in the revision is a positive step toward reproducibility.
> > 2: The clarification regarding Table 3 is crucial—knowing that all baselines were fine-tuned on the VPD-100K dataset ensures a fair comparison. The 4.8% improvement attributed to the FEM module is now more clearly understood as a structural gain rather than a data advantage. Additionally, providing the specific hardware (A100) and resolution (1920x1088) details for the latency benchmarks in Table 4 resolves the previous ambiguity.
> > 3: While the responses successfully defend the paper's current claims, the core contribution remains an incremental though solid improvement (FEM module) on top of existing YOLO architectures for a specific application (privacy detection).
> > Summary:
> > The rebuttal successfully addressed the technical and ethical questions raised in the initial review. The paper is well-executed and the dataset VPD-100K appears to be a valuable contribution to the community. I maintain my original score and recommend acceptance based on the clarified experimental results.

---

### Official Review · Reviewer_ZhPN · 2026-03-12

**Soundness:** 3
**Presentation:** 3
**Significance:** 4
**Originality:** 4
**Overall Recommendation:** 5
**Confidence:** 4

**Summary:**

This paper provides a foundational contribution to the field of visual privacy. The release of a high-quality, 100k-image dataset with fine-grained annotations is a major service to the research community. Furthermore, the technical integration of frequency-domain processing into modern real-time detectors provides a clear performance boost for the specific challenges of privacy detection (small/blurry text and faces) without sacrificing speed. The comprehensive evaluation and ethical rigor make it a standout submission.

**Compliance With Llm Reviewing Policy:**

Affirmed.

**Final Justification:**

The authors have fully addressed my concerns.

**Key Questions For Authors:**

1.	The paper notes that the dataset is available, which is vital for the community
2.	While the user study with 20 participants is a good starting point, future iterations could benefit from a larger, more diverse demographic to further validate the "Comfort" and "Adoption" metrics across different cultures
3.	The balance hyperparameter β=0.05 is well-justified to prevent high-frequency gradients from overwhelming the primary task, showing deep understanding of the optimization process

**Limitations:**

please see the weaknesses.

**Strengths And Weaknesses:**

N/A

---

> ### Author Rebuttal · Authors · 2026-03-31
>
> We greatly appreciate your insightful and detailed feedback. Your comments have been carefully discussed and fully taken into account, and we improve our work accordingly.
>
> > **1.Dataset Availability**:
>
> We sincerely thank the reviewer for recognizing the value of our contribution. As promised, we are fully committed to open-sourcing the complete VPD-100K dataset upon publication. We hope this dataset and benchmark will serve as a foundational resource to drive future research and facilitate the development of robust visual privacy protection systems.
>
> > **2.Usability Evaluation**:
>
> We agree that expanding the user study to include a larger and more diverse demographic, particularly across different cultural backgrounds, would further strengthen the evaluation of subjective dimensions such as Comfort and Adoption.
>
> In the current work, our user study is designed as a "light-weight" usability evaluation to validate the completeness of the proposed privacy taxonomy and the general perceived usefulness of the system, rather than to perform a large-scale statistical study aiming for population-level inference.
>
> To improve robustness, we have expanded the participant pool from 20 to 40 users, and updated results consistently show strong positive agreement across all dimensions, where 5 represents Strong Accept.
>
> We will revise the manuscript to explicitly highlight cross-cultural evaluation as an important direction for future work.
>
> | Rating of Solution | N=20 Mean (±SD)| N=40 Mean (±SD) |
> |---|----|---|
> | Comfort| 4.35 ± 1.04 | 4.25 ± 0.84 |
> | Relevance| 4.40 ± 0.75 | 4.28 ± 0.69 |
> | Protection| 4.50 ± 0.61| 4.33 ± 0.57|
> | Usefulness| 4.30 ± 0.92| 4.25 ± 0.71|
> | Trust| 4.20 ± 0.95 | 4.20 ± 0.72 |
> | Reliefness| 4.15 ± 0.88 | 4.18 ± 0.68|
> | Adoption|4.05 ± 0.94| 4.08 ± 0.80|
> | Recommendation| 4.20 ± 0.83| 4.20 ± 0.65|
>
> > **3.Sensitivity analysis**:
>
> We sincerely thank for your highly encouraging feedback. Carefully balancing the high-frequency boundary refinement with the primary spatial detection convergence is a core optimization challenge in our design. We also conduct additional experiments on the VPD-100K test dataset using YOLOv10-S+our FEM as the baseline, sweeping β across a range of values. The results are summarized below, and it also supports our understanding of the optimization process.
> | Method ($\beta$ Value) | AP ↑ | AP₅₀ ↑ | AP₇₅ ↑ | APₛ ↑ | F1-Score ↑ |
> | :--- | :---: | :---: | :---: | :---: | :---: |
> | + $\beta = 0.01$ | 50.9 | 65.8 | 53.9 | 29.2 | 0.68 |
> | **+ $\beta = 0.05$ (Ours)**| **52.1** | **67.1** | **54.6** | **30.1** | **0.71** |
> | + $\beta = 0.10$ | 51.1| 66.2 | 54.2 | 29.6 | 0.69 |

---

> > ### Author Rebuttal · Reviewer_ZhPN · 2026-04-01
> >
> > The authors have fully addressed my concerns.

---

### Official Review · Reviewer_f2Cr · 2026-03-12

**Soundness:** 2
**Presentation:** 2
**Significance:** 2
**Originality:** 2
**Overall Recommendation:** 3
**Confidence:** 3

**Summary:**

The paper introduces VPD-100K, a large-scale, fine-grained visual privacy dataset (100,000 images across four domains) that addresses critical gaps in existing datasets scale, taxonomy, and domain coverage. Furthermore, they propose and validate a lightweight Frequency-Enhanced Mechanism (FEM), which integrates frequency-domain attention fusion, adaptive spectral gating, and a frequency-consistency loss,verifying its effectiveness on the VPD-100K.

**Compliance With Llm Reviewing Policy:**

Affirmed.

**Final Justification:**

I appreciate the authors' response. However, regarding the question of why newer models like YOLOv11-s and YOLOv12-s yield worse results than the proposed method based on YOLOv10-s, The authors' explanation is not entirely convincing. I remain concerned that the method is specifically over-tuned for YOLOv10-s, as its poor performance on newer frameworks exposes a lack of generalizability. Therefore, I maintain my original score.

**Key Questions For Authors:**

see Strengths And Weaknesses

**Strengths And Weaknesses:**

Despite its merits, the paper presents the following shortcomings:

(1) Fails to include current SOTA models like YOLOv11 and YOLOv12 for accuracy-latency comparisons.

(2) As a plug-and-play module, FEM is not compared against standard attention mechanisms (e.g., CBAM, SENet, Mamba) to justify its superiority.

(3) Lacks hyperparameter sensitivity analysis (e.g., for $\beta$).

(4) Unclear how the dataset anonymizes unintended background subjects (e.g., bystanders, license plates) in real-world video streams to prevent secondary privacy leaks.

(5) The paper claims to have conducted a comprehensive user study, yet it only involves 20 participants. Such a small sample size is statistically insufficient to draw robust conclusions.

(6) While the proposed method is intuitive and easy to implement, it lacks an in-depth theoretical analysis of the underlying mechanisms driving its effectiveness. The paper does not provide the code.

---

> ### Author Rebuttal · Authors · 2026-03-31
>
> We greatly appreciate your insightful and detailed feedback.
> > **1.More SOTA Accuracy-latency comparisons**:
>
> We compare our method with the most recent baselines YOLOv11 and YOLOv12 to demonstrate the superiority of our mechanism as follows:
> |Base|$AP^{V}$| $AP_{50}^{V}$|$AP_{75}^{V}$|$AP_{s}^{val}$| $AP_{M}^{V}$ |$AP_{L}^{V}$ |GFLPOS|Latency(ms)| F1-Score
> | :--- | :---: | :---: | :---: | :---:| :---:| :---:| :---:| :---:| :---:|
> |YOLOv11-s|46.5|63.5| 51.2|26.5|53.7| 63.3|22.9| 2.55| 0.67|
> |YOLOv12-s|47.8|64.9|52.5|27.1 |54.8|**64.5**|23.1| 2.60| 0.66|
> |Our-s|**52.1**|**67.1**|**54.6**|**30.1**|**55.6**|64.3|26.0|2.71|**0.71**|
>
> > **2.Attention analysis**:
>
> We conduct  experiments by replacing our FEM with SENet, CBAM attention within the same baseline.
> It indicates that our FEM achieves more favorable results with similar computational overhead. The experiments are evaluated under identical configurations in the table below:
> |Base |$AP^{V}$|$AP_{50}^{V}$|$AP_{75}^{V}$|$AP_{s}^{val}$ |$AP_{M}^{V}$|$AP_{L}^{V}$|GFLPOS|Latency(ms)| F1-Score
> |:---|:---:|:---:|:---:|:---:| :---: | :---: | :---: | :---: | :---: |
> |SENet|47.1|63.4|52.1|26.5|54.0|63.2|23.1|2.59|0.66|
> |CBAM|47.8|64.1|52.4|27.2|54.8|63.9|23.4|2.61| 0.68|
> |ours FEM|**52.1**|**67.1**|**54.6**|**30.1**|**55.6**|**64.3**|26.0|2.71|**0.71**|
>
> > **3.Sensitivity analysis**:
>
> We conduct additional experiments on the VPD-100K test set using YOLOv10-S+our FEM as the baseline, sweeping β across a range of values. The results are summarized to show the superiority of our setting below:
> | Method ($\beta$ Value) | AP ↑ | AP₅₀ ↑ | AP₇₅ ↑ | APₛ ↑ | F1-Score ↑ |
> | :--- | :---: | :---: | :---: | :---: | :---:|
> | + $\beta = 0.01$ | 50.9 | 65.8 | 53.9 | 29.2 | 0.68 |
> | **+ $\beta = 0.05$ (Ours)**|**52.1**| **67.1**| **54.6**|**30.1**|**0.71**|
> | + $\beta = 0.10$ |51.1|66.2| 54.2| 29.6|0.69|
>
> > **4.Eliminating secondary privacy risk analysis**:
>
> Since VPD-100K trains defensive models, pre-anonymizing targets would destroy essential detection signals.
> To mitigate privacy risks, all data is ethically sourced from public or simulated environments with Ethical Review Board approval.
> Furthermore, we provide only categorical annotations to structurally prevent individual re-identification.
> The dataset will be released under a strict research-only license, explicitly prohibiting malicious applications like surveillance.
>
> > **5.Sample size of the user study**:
>
> Our study is designed as a "light-weight" usability evaluation to validate the privacy taxonomy completeness and the alignment between model and human.
> To strengthen empirical validity, we add 20 participants, totaling 40, matching prior human computer interaction studies [1, 2].
> Updated results consistently show positive agreement across all dimensions, where 5 is Strong Accept.
>
> | Rating of Solution | N=20 Mean (±SD)| N=40 Mean (±SD) |
> |---|----|---|
> | Comfort| 4.35 ± 1.04 | 4.25 ± 0.84 |
> | Relevance| 4.40 ± 0.75 | 4.28 ± 0.69 |
> | Protection| 4.50 ± 0.61| 4.33 ± 0.57|
> | Usefulness| 4.30 ± 0.92| 4.25 ± 0.71|
> | Trust| 4.20 ± 0.95 | 4.20 ± 0.72 |
> | Reliefness| 4.15 ± 0.88 | 4.18 ± 0.68|
> | Adoption|4.05 ± 0.94| 4.08 ± 0.80|
> | Recommendation| 4.20 ± 0.83| 4.20 ± 0.65|
>
> | Privacy Instance | N=20 Mean (±SD) | N=40 Mean (±SD) |
> |---|---|---|
> | Human| 4.30 ± 0.98 | 4.25 ± 0.78 |
> | On-screen PII| 4.40 ± 0.88  | 4.30 ± 0.69 |
> | Physical Identifiers | 4.45 ± 0.69 | 4.38 ± 0.59 |
> | Location | 4.45 ± 0.76 | 4.33 ± 0.66 |
> | Total |4.50 ± 0.69| 4.40 ± 0.63 |
>
> [1]ImageAlly: A Human-AI hybrid approach to support blind people in detecting and redacting private image content. SOUPS 2023.
>
> [2]U-VAP: User-specified visual appearance personalization via decoupled self augmentation. CVPR 2024.
>
> > **6.Theoretical analysis and Code Availability**:
>
> We theoretically justify the efficacy of our Frequency-Enhanced Mechanism for privacy detection.
>
> **Overcoming the Low-Pass Bias of Spatial Convolutions**. CNNs and spatial modules exhibit a known spectral bias towards low-frequency information. However, critical privacy instances (like tiny verification codes) are inherently rich in high-frequency details. Our FDAF explicitly maps features to the frequency domain to bypass this spatial bias, allowing the network to directly access and manipulate high-frequency components without spatial loss.
>
> **Mechanism of Adaptive Spectral Gating**. The learnable mask W-gate acts as a dynamic band-pass filter. For structured privacy objects (e.g., text), the optimization process forces W-gate to exhibit strong activations in specific directional frequency bands.
>
> **Spectral Alignment**: Our Frequency-Consistency Loss ($\mathcal{L}_{freq}$) follows the principle of maintaining spectral power consistency between predictions and targets. By explicitly minimizing frequency-domain distance, we force the model to prioritize high-frequency boundary information, crucial for localizing tiny privacy targets.
>
> The code and the dataset will be publicly released.

---

> > ### Author Rebuttal · Reviewer_f2Cr · 2026-04-03
> >
> > I thank the authors for their response. However, I remain concerned about the counterintuitive performance drops in the baselines. Specifically, why do newer models like YOLOv11-s and YOLOv12-s yield worse $AP^{V}$ than your proposed method based on YOLOv10-s? Furthermore, how would the performance compare if evaluated against a larger-capacity model, such as YOLOv12-L? Could the authors elaborate on the underlying reasons for these observations? I am also concerned about whether the hyperparameters for these newer models were sufficiently tuned to ensure a fair comparison. Finally, while the authors have provided extensive empirical results, the rebutal still lacks an in-depth analysis to explain the mechanisms behind these phenomena.

---

> > > ### Author Response · Authors · 2026-04-04
> > >
> > > Thanks for your detailed feedback.
> > >
> > > > **1.Clarification on "Ours-s" & YOLOv11/12-L Baselines**:
> > >
> > > We clarify that "Ours-s" does not represent the native YOLOv10-s, it denotes our proposed method: the **YOLOv10-s baseline equipped with our Frequency-Enhanced Mechanism (FEM)**.
> > >
> > > As you expected, YOLOv11s/12s outperform native YOLOv10-s (at Tab. 3 of our manuscript). For fairness, all baselines strictly use official default hyperparameters.
> > >
> > > Crucially, equipping the weaker YOLOv10-s with FEM (denoted as "Ours-s" ) yields an 8.9\% AP(V) gain over the strongest small baseline (YOLOv12-s). This proves FEM's impact significantly surpasses mere network upgrades.
> > >
> > > Additionally, new experiments verify that FEM successfully generalizes to larger models (YOLOv11-L and YOLOv12-L) as follows:
> > >
> > > |Base|$AP^{V}$| $AP_{50}^{V}$|$AP_{75}^{V}$|$AP_{s}^{val}$| $AP_{M}^{V}$ |$AP_{L}^{V}$ |GFLPOS|Latency(ms)| F1-Score
> > > | :--- | :---: | :---: | :---: | :---:| :---:| :---:| :---:| :---:| :---:|
> > > |YOLOv11-L|53.9|69.7|58.4|33.8|60.1|70.9|88.2|6.45|0.74|
> > > |YOLOv12-L|54.2|70.1|58.9|34.3|60.4|71.3|90.4|6.91|0.74|
> > > |YOLOv11-L+our FEM|58.7|73.4|61.4|36.7|62.4|70.7|99.6|6.48|0.81|
> > > |YOLOv12-L+our FEM|59.1|74.0|61.8|37.0|62.9|71.1|101.6|7.06|0.83|
> > >
> > > > **2.Theoretical analysis**:
> > >
> > > Our Adaptive Spectral Gating implicitly constructs a filter in the frequency domain by learning process. This filter aims to minimize the discrepancy between the filtered input features and the signals capturing crucial high-frequency details. Formally, we define our input as
> > > $$ X=S+N $$
> > > where $S$ represents the high-frequency information and $N$ represents the noises. We first transfer it to frequency domain through DFT and obtains
> > >
> > > $$(1) \mathbf{F}_X = \mathbf{F}_S + \mathbf{F}_N$$
> > >
> > > because DFT is a linear transformation. Our design aims to implicitly solve the following objective:
> > >
> > > $$(2)\min_{\mathbf{m}} \mathbb{E} \left[ || \mathbf{m} \odot\mathbf{F}_X - \mathbf{F}_S ||_2^2 \right].$$
> > >
> > > Let $\mathbf{F}_X, \mathbf{F}_S, \mathbf{F}_N \in \mathbb{C}^{D}$,  $\mathbf{M} = \text{diag}(\mathbf{m}) \in \mathbb{R}^{D \times D}$ be the diagonal form of the filter, where the learnable one is $\mathbf{m} \in (0,1)^D$. Substituting (1) into the (2), we transfer the objective to
> > >
> > > $$\min_{\mathbf{m}} J(\mathbf{m}) = \mathbb{E} \left[ || (\mathbf{M} - \mathbf{I}) \mathbf{F}_S + \mathbf{M} \mathbf{F}_N ||_2^2 \right].$$
> > >
> > > Under assumptions that the signal and the noise are uncorrelated, their covariances vanish as $\mathbb{E}[\mathbf{F}_S \mathbf{F}_N^H] = \mathbf{0}$ and $\mathbb{E}[\mathbf{F}_N \mathbf{F}_S^H] = \mathbf{0}$, where $H$ denotes the Hermitian transpose. The objective then naturally decouples into the signal distortion and the residual noise:
> > >
> > > $$J(\mathbf{m}) = \mathbb{E} \left[ \mathbf{F}_S^H (\mathbf{M} - \mathbf{I})^2 \mathbf{F}_S \right] + \mathbb{E} \left[ \mathbf{F}_N^H \mathbf{M}^2 \mathbf{F}_N \right].$$
> > >
> > > We define the power spectral density for the signal and noise as $\mathbf{\Sigma}_S = \text{diag}(\mathbb{E}[|\mathbf{F}_S|^2])$ and $\mathbf{\Sigma}_N = \text{diag}(\mathbb{E}[|\mathbf{F}_N|^2])$. Then the objective function is rewritten in $\mathbf{m}$:
> > >
> > > $$J(\mathbf{m}) = (\mathbf{m} - \mathbf{1})^T \mathbf{\Sigma}_S (\mathbf{m} - \mathbf{1}) + \mathbf{m}^T \mathbf{\Sigma}_N \mathbf{m}.$$
> > >
> > > To obtain the solution $\mathbf{m}^*$, we take first-order condition and obtain
> > >
> > > $$\nabla_{\mathbf{m}} J = 2 \mathbf{\Sigma}_S (\mathbf{m} - \mathbf{1}) + 2 \mathbf{\Sigma}_N \mathbf{m} = \mathbf{0}.$$
> > >
> > > It provides the theoretical global minimum:
> > >
> > > $$\mathbf{m}^* = (\mathbf{\Sigma}_S + \mathbf{\Sigma}_N)^{-1} \mathbf{\Sigma}_S \mathbf{1}.$$
> > >
> > > The results represent a Wiener Filter. However, in real-world visual tasks, explicitly acquiring the exact PSD is intractable.
> > >
> > > We note that the elements of $\mathbf{m}^*$ inherently reside within the interval $(0, 1)$, and the magnitude of
> > >
> > > $\mathbf{m}^*$ is positively correlated with the energy proportion of the target signal.
> > >
> > > Therefore, we design a learning-based paradigm, introducing the Adaptive Spectral Gating Mechanism to adaptively approximate the filter using a simple network, where the activation function ensures the output values remain in $(0, 1)$, aligning with the theoretical constraints.
> > >
> > > Moreover, inspired by this positive correlation between the magnitude and the high-frequency signal energy ratio, we design the frequency-consistency loss $\mathcal{L}_{freq}$ to explicitly amplify this effect. Ultimately, our method can be interpreted as a neural approximation
> > >
> > > $$ \sigma(\mathbf{W}_{gate}) \to \mathbf{m}^*,$$
> > >
> > > which implicitly minimizes the expected discrepancy between the filtered signal and the target signal, with effective performance in practical tasks.
> > >
> > > **We hope these updates resolve your concerns and improve your assessment.**

---

### Decision · Program_Chairs · 2026-04-30

**Decision:**

Accept (regular)

**Comment:**

The paper received two accept recommendations and one weak reject recommendation from reviewers. Reviewers recognized the VPD-100K dataset as a valuable contribution to the community, highlighting its large scale, fine-grained annotations, and practical importance for visual privacy protection. Reviewers also found the proposed frequency-enhanced mechanism effective in improving detection performance, although one reviewer viewed the methodological contribution as more incremental and raised concerns about generalizability across newer YOLO backbones. After rebuttal, the two positive reviewers indicated that their concerns had been fully addressed, while the remaining reviewer maintained a weak reject. After considering the paper, reviews, and rebuttal, AC believes that the overall contribution of the dataset resource and technical approach outweighs the remaining concerns and recommends acceptance. Authors are encouraged to incorporate the clarifications on data collection, experimental settings, and baseline discussion into the final version.